# EcoLight: Intersection Control in Developing Regions Under Extreme Budget and Network Constraints

**Sachin Kumar Chauhan, Kashish Bansal and Rijurekha Sen**
Department of Computer Science and Engineering
Indian Institute of Technology Delhi
{csz188012,cs5150285,riju}@cse.iitd.ac.in

## Abstract

Effective intersection control can play an important role in reducing traffic congestion and associated vehicular emissions. This is vitally needed in developing countries, where air pollution is reaching life threatening levels. This paper presents EcoLight intersection control for developing regions, where budget is constrained and network connectivity is very poor. EcoLight learns effective control offline using state-of-the-art Deep Reinforcement Learning methods, but deploys highly efficient runtime control algorithms on low cost embedded devices that work standalone on road without server connectivity. EcoLight optimizes both average case and worst case values of throughput, travel time and other metrics, as evaluated on open-source datasets from New York and on a custom developing region dataset.

## 1 Introduction

Developing countries are overwhelmed with the problems of traffic congestion (TimesOfIndia [2018], IndiaTimes [2018], FinancialExpress [2018]) and air pollution (DW [2019a], Amnesty [2019], DW [2019b]). Intersection control to better manage traffic congestion and reduce vehicular emissions is vitally needed, in addition to policies for curbing traffic (Ecotech [2016]). State-of-the-art intelligent intersection control like Presslight (Wei et al. [2019a]) and CoLight (Wei et al. [2019b]) are showing impressive results on lane-based orderly traffic of the developed countries. This paper explores whether the benefits of these Convolutional Neural Network (CNN) based real time video analysis from traffic cameras, and Deep Reinforcement Learning (DRL) based adaptive intersection control, can be translated to the developing world, where the need for these technologies is paramount.

The challenges of directly importing the afore-mentioned technologies are three fold – (a) extreme budget constraints, which allows for only very low cost, compute and RAM constrained, embedded platforms to be deployed (b) poor network connectivity between the road and the servers, forcing all analysis to happen in-situ on the road and (c) chaotic non-laned driving behavior in developing regions, which makes accurate video analysis for exact counting and classification of vehicles harder.

This paper presents EcoLight, the first practical step towards intelligent intersection control in developing countries. Through close collaboration with traffic authorities and cameras deployed in real intersections, we explore how low computation intensive video processing and control algorithms can be trained to run on low cost embedded devices without network connectivity. EcoLight exploits state-of-the-art CNN and DRL methods on high-end GPUs in the pre-deployment stage – (a) CNN for training more efficient traffic density estimation using background subtraction and optical flow, and (b) DRL for learning efficient Look-Up Table (LUT) based or threshold based intersection control. This enables EcoLight to perform at par with these compute intensive methods, at a mere fraction of runtime overhead.

Optimizing computational overhead while not losing accuracy has been challenging for EcoLight. We reduce DRL states from over a thousand dimensions in state-of-the-art papers (Wei et al. [2019a,b]) to one or two dimensions. We remove the DNN based RL computation at runtime using static LUTs. We quantize the original continuous values of DRL states for finite sized LUTs. All these optimizations needed to be carefully tuned for accuracy. We experiment with both open-source developed country dataset and a custom developing region dataset, created by us from our deployed cameras. As a result of careful tuning, EcoLight gives comparable benefits and sometimes even improves upon the compute-intensive methods, on both performance metrics (throughput, average travel time etc. at the intersection) and fairness metrics (worst case travel time, vehicles stuck etc. at the intersection).

An end-to-end EcoLight based system has also been demonstrated in this paper. This incorporates video feeds from cameras at a real intersection and computer vision based traffic density estimation for input to the control algorithms. Our results show great promise towards practical adaptive intersection control at extreme budget and network constraints, a vital necessity for sustainability.

## 2  Problem Definition

To start-with, we define the problem of traffic signal control as a Markov Process. Each intersection in the system is controlled by an agent running independently, and without any communication with the others. In this setting, each agent observes part of the total system, and decides for its own intersection whether to keep the same phase or switch to the next, so as to minimize the average traffic density on the approaches around the intersection. Specifically, the problem can be characterized by the following major components $< \mathcal{S}, \mathcal{O}, \mathcal{A}, \mathcal{P}, r, \pi, \gamma >$ as described in detail below.

• With system state space $\mathcal{S}$ and observation space $\mathcal{O}$, we assume that there are $N$ intersections in the system and each agent can observe part of the system state $s \epsilon \mathcal{S}$ as its observation $o \epsilon \mathcal{O}$. We define $o_i^t$ for agent $i$ at time $t$, which consists of traffic density in one or two dimensions as described later.

• With set of actions $\mathcal{A}$, at time $t$, an agent $i$ would choose an action $a_i^t$ from its candidate action set $\mathcal{A}_i$ as a decision for the next $\Delta t$ period of time. Here, each agent would choose either 0 or 1 as its action $a_i^t$, indicating that from time $t$ to $t + \Delta t$, this intersection would be in same phase or under transition to the next phase.

• With transition probability $\mathcal{P}$, given the system state $s_i^t$ and actions $a_i^t$ of agent $i$ at time $t$, the system arrives at the next state $s_i^{t+1}$ according to the state transition probability $P(s_i^{t+1}|s_i^t, a_i^t)$.

• With reward $r$, each agent $i$ obtains an immediate reward $r_i^t$ from the environment at time $t$. In this paper, we want to minimize the travel time for all vehicles in the system, which is hard to optimize directly. Therefore, we define the reward for intersection $i$ as $r_i^t = -\sum_a d_{i,a}^t$ where $d_{i,a}^t$ is the stop density on the approach $a$ of intersection $i$ at time $t$.

• With Policy $\pi$ and discount factor $\gamma$, as the independent actions have long-term effects on the system, we want to minimize the expected stop density of each intersection in each episode. Specifically, at time $t$, each agent chooses an action following a certain policy $\mathcal{O} \times \mathcal{A} \to \pi$, aiming to maximize its total reward $G_i^t = \sum_{t=\tau}^T \gamma^{t-\tau} r_i^t$, where $T$ is total time steps of an episode and $\gamma \epsilon [0, 1]$ differentiates the rewards in terms of temporal proximity.

In this paper, we use the action-value function $Q_i(\theta_n)$ for each agent $i$ at the $n^{th}$ iteration (parameterized by $\theta$) to approximate total reward $G_i^t$ with neural networks by minimizing the loss:

$$\mathcal{L}(\theta_n) = E[(r_i^t + \gamma \max_{a'} Q(o_i^{t'}, a_i^{t'}; \theta_{n-1}) - Q(o_i^t, a_i^t; \theta_n))^2] \qquad (1)$$

where $o_i^{t'}$ denotes the next observation for $o_i^t$. These earlier snapshots of parameters are periodically updated with the most recent network weights and help increase the learning stability by de-correlating predicted and target q-values.

## 3  Doing Away with Large States: Small State DRL for Intersection Control

Deep Reinforcement Learning (DRL) based intersection control algorithms like Presslight (Wei et al. [2019a]), CoLight (Wei et al. [2019b]) are giving excellent performance in recent literature. Unfortunately the DRL state size for these state-of-the-art algorithms are 80 for Presslight and 1600-12480 for CoLight. Further Presslight needs coordination among different intersections, though it learns individual RL agents for each intersection. CoLight learns a centralized RL agent. Coordinated or centralized control requires network connectivity, which is not ubiquitous in developing regions.

For deployment on low cost embedded devices (with limited RAM and compute power) and per intersection control without coordination, we start with exploring small state DRLs. Let $x1$ denote the traffic density on the road approach with green signal and $x2$ denote total traffic density on all other road approaches with red signal. We explore two small state DRLs - (a) 2-dimensional state $< x1, x2 >$ (b) 1-dimensional state $< x3 >$, where $x3 = x1/(x1 + x2)$.

Table 1: Performance of 2-dimensional and 1-dimensional state RL

| Model | DNN | | 1x1 (3-approach) 123 veh/min @ Delhi | | | 16x1 (4-approach) 114 veh/min @ NY | | | 16x3 (4-approach) 47 veh/min @ NY | | |
|---|---|---|---|---|---|---|---|---|---|---|---|
| | StateSz | Param | nOut | Travl | Total | nOut | Travl | Total | nOut | Travl | Total |
| Presslight | 80 | 2082 | 1246 | 254 | 252 | 4866 | 220 | 362 | 1355 | 560 | 930 |
| CoLight | 12480 | 6018 | 1248 | 222 | 251 | 4986 | 260 | 375 | 2589 | 319 | 311 |
| 2dimRL | 2 | 162 | 1282 | 238 | 243 | 5010 | 252 | 376 | 2574 | 328 | 322 |
| 1dimRL | 1 | 52 | 1260 | 254 | 250 | 3607 | 187 | 604 | 1148 | 375 | 1093 |

Using (a) 1-hour long publicly available New York (USA) dataset[1] and (b) 1-hour New Delhi (India) dataset collected and processed by us, we simulate the traffic-flows in the CityFlow traffic simulator (Zhang et al. [2019]). Table 1 show the metric values that need to be maximized (nOut or number of vehicles cleared by the controller) and minimized (Travel and Total times). Total combines the time spent by the vehicles which clear the intersection and also those stuck at the intersection, while Travel time comprises only cleared vehicles' time spent in the network.

We evaluate two road networks in the New York dataset – (a) 16x3 network, where 16 roads parallel to each other intersect with 3 roads perpendicular to them giving 48 intersections, each having 4 approaches, and (b) 16x1 network with one perpendicular road to 16 parallel roads, giving 16 intersections, each with 4 approaches. The Delhi dataset is for one intersection (1x1), with 3 approach.

As seen from the table, the 1-dimensional state DRL does poorly on Throughput and TotalTime metrics, especially on 16x3 network. However, the 2-dimensional state DRL shows impressive metric values, matching the performance of CoLight and improving upon Presslight, at a mere fraction of state size and parameters. This shows that the state-of-the-art DRL algorithms have a lot of redundancy, that can be optimized, and also less DNN parameters for the small state DRLs can be trained better with limited data. This result shows the promise of small state DRLs. We take this as the starting point to build our efficient intersection control in the next two sections.

## 4   Doing Away with Runtime DRL: Lookup Table based Intersection Control

In Section 3, we use a DRL architecture with fully connected layers, comprising two hidden layers of size $H$ each. We use $M$ dimensional states to represent an intersection and further show in Section 3, that $M = 2$ gives comparable results to state of the art intersection control algorithms Presslight and CoLight (Wei et al. [2019a,b]). We consider $N$ phases, each phase denoting a certain configuration of green and red signals for the different approaches at the intersection. At every decision making point, our DRL can make one of two choices, to stay in the current phase or switch to the next phase. So our DRL has an $M$ x $H$ x $H$ x $P$ architecture, with $M = 2$, $H = 10$ and $P = 2$. We use stop density as the DRL reward, which is easily computable as discussed later in section 6. Though our optimizations and method should improve the performance irrespective of underlying Loss Function and Optimizer choice, still we used them same as the baselines, MeanSquareError (Verma [2019]) and RMSprop (Bushaev [2018]), to rule out their effect from the performance comparison/improvement.

In this section, we seek to do away with running the DRL at runtime at the deployment site. The first reason is efficiency: on low cost embedded systems, compute power is limited. The inputs for the control algorithms anyway needs to be computed on the embedded devices, using computer vision algorithms on the real time video data from all approaches. Using these inputs, if the control algorithm can be made more efficient than running a neural network for DRL, it becomes more practical to meet the low computational budget. The second reason to do away with runtime DRL, is the lack of confidence on the DRL black box. Based on anecdotal evidence through discussions with our deployment partners, adaptive intersection control that can be visualized and verified by human experts before deployment, is much more preferred than algorithms which are free to choose actions at runtime without any human supervision/comprehension, as a runtime DRL would do.

We therefore seek to use static Lookup Tables (LUT) at deployment, where each cell in the table will represent a state in our DRL. The value contained in that cell will represent a boolean action: stay in the current phase vs. switch to the next phase, referred to as *keep-change* actions henceforth. The actions are learnt using offline DRL training. This training can be compute heavy and high latency, as it is run on powerful GPU servers before deployment for real time intersection control. During training, computer vision based processed video datasets are collected from the road, and fed in traffic simulator to create all possible DRL states (cells in the LUT). Actions corresponding to each state are then learnt by training the DRL algorithm.

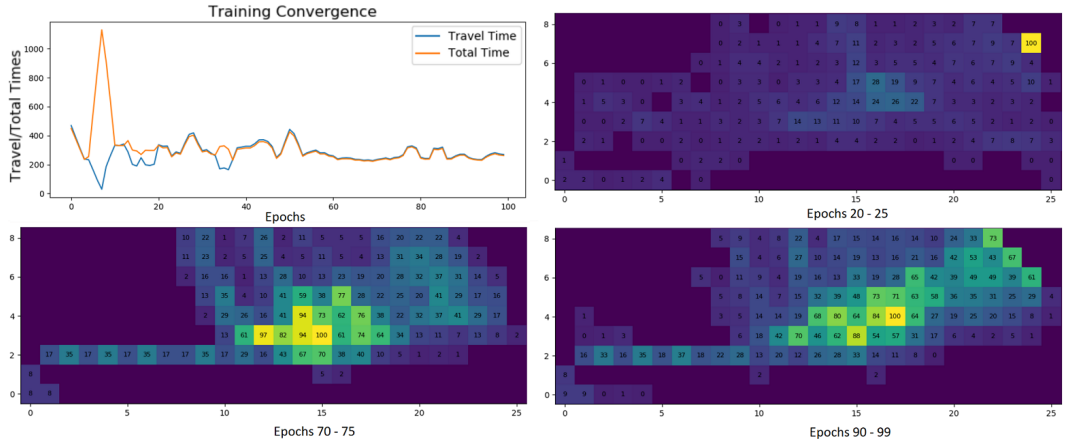

Figure 1: DRL training and LUT structure

The first graph in Figure 1 shows how metrics Total time and Travel time improve over many epochs of offline DRL training. The other three images show how many times different DRL states are seen by the DRL training algorithm as training progresses. The lighter the color, the more a DRL state is seen. These three images also describe the LUT structure, where the two axes represent quantized values of $x1$ and $x2$ for the 2-dimensional state DRL. Instead of "how many times a DRL state is seen" presented in these images, the LUT contains a boolean action value in each cell, learnt by DRL training. Verified by developing country traffic control experts for sanity and safety checks, the LUT is eventually deployed on road. At runtime, the current state is computed using computer vision methods on incoming video, and the action corresponding to that state in the stored LUT is taken by the traffic signal controller.

While storing DRL decisions for different states in LUT is efficient and verifiable, we need to ensure that the learnt decisions are good for subsequent use at runtime. It is important to choose good DRL models to populate the static LUT, as unlike running DRL at runtime, the LUT will not be able to dynamically update these decisions.

As measure of DRL model goodness, we define two metrics:
**(a) FairShare:** We hypothesize that a good RL tries to achieve FairShare of traffic densities among approaches i.e. fit the traffic among at the intersection such that each approach maintains equal/similar density of traffic. To quantify this FairShare property of a given DRL model, we project all instances of observed states (factored by the distance) onto the equal density segments of LUT (corresponding to the diagonal starting at 0,0) in Figure 1. We sum this vector of the projections to get a single scalar, which will be high for models with most states with equal density (like Epoch 90-99 in Figure 1), and low otherwise. This scalar quantifies how balanced traffic is among the approaches for a particular DRL model.

**(b) DecisionConsistency:** If a model predicts to hold/keep the signal for a state, we hypothesize that a good or stable model should continue to predict the same for all states having higher traffic in the green approach (or low traffic in the red approaches). We name this model property of sticking to the same decision under similar traffic scenarios as DecisionConsistency. To quantify DecisionConsistency, for each green density level ($x1$) we take the ratio of two numbers, the large range of red density ($x2$) over which the keep decision is maintained vs the range followed with opposite decision. The sum of all such ratios gives rise to a scalar which will be larger for models with better DecisionConsistency.

In addition to hypothesizing what properties good DRL models might have, and defining scalar metrics to quantify those goodness properties, we also need mechanisms to use these goodness metrics. We do this in the following two ways:

**(a) DRL training using model goodness metrics:** We use the FairShare and DecisionConsistency scalars during the DRL training process to identify and favour better RL models. We maintain a threshold $\theta$ for these scalars, as training progresses. At each epoch we hold a model if its goodness metric is below $\theta$, lower $\theta$ by a factor, and start the training for a fresh model in that epoch. We approve the best model so far (new or on hold), if its goodness metric exceeds $\theta$, or after fixed number ($\eta$=5) of retries in that epoch, and move on with the metric value of this model as new $\theta$.

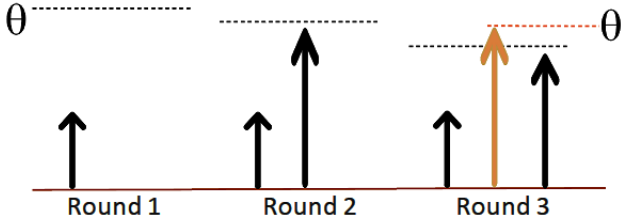

Figure 2: Goodness metrics based DRL training

**(b) DRL selection using model goodness metrics:** Figure 3 shows the correlation between Total Time performance metric and DRL model's goodness metric values. We discard models with goodness metric values lower than the average of all the models, to remove outliers (see Perspective 1 of Figure 3). In order to select the good models among the remaining ones, we pick the best model (again based on the goodness metric values) among a set of ($\psi$=20) models, and restart the process from the model next to the selected one (see Perspective 2 of Figure 3). This final set of high performing models can be effectively used to generate the LUT to be deployed at the intersection.

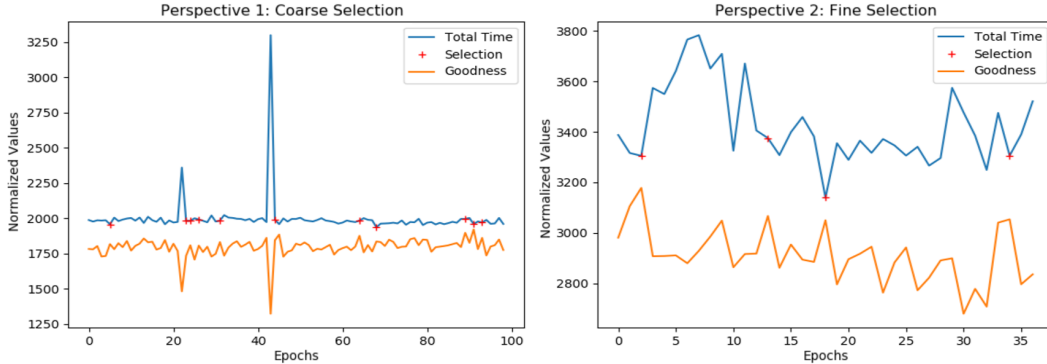

Figure 3: Goodness metrics based DRL selection

We need to evaluate this LUT based signal control, compared to the 2dimRL that we designed in Section 3, and also the state-of-art DRL methods Presslight (Wei et al. [2019a]) and CoLight (Wei et al. [2019b]). Static LUTs lose performance due to quantization of the traffic density values, while runtime DRL can use continuous values of traffic density. But the quantization is unavoidable, as the table needs to be of finite dimensions. Whether our training and training+selection with goodness metrics can overcome the quantization related performance loss, needs to be quantified.

Table 2 shows the average case performance metric values (a) nOut (number of vehicles cleared by the intersection), (b) Travel (time spent by cleared vehicles) and (c) Total (time spent by all vehicles). The T in model names denotes Goodness based Training only experiments, whereas TS includes Goodness based Selection as well. We continue the training for 200 epochs, allowing all methods to converge and then average the next 50 epochs for performance metrics calculation for T, and the selected few out of these for TS. As can be seen from the table, performance loss compared to 2dimRL due to quantization, is gracefully recovered by both our goodness metrics. DecisionConsistency performs significantly better than FairShare for all datasets.

We further show the value of worst case or fairness metrics for 16x3 benchmark dataset in Table 3. Our fairness metrics are: (a) WrstTime (maximum time spent in the network by any stuck vehicle), (b) WrstWait (maximum wait time at any intersection by any vehicle), (c) MaxWait (maximum of

Table 2: Performance of Goodness EcoLight for average case metrics

| Model | 1x1 | | | 16x1 | | | 16x3 | | |
|---|---|---|---|---|---|---|---|---|---|
| | nOut | Travel | Total | nOut | Travel | Total | nOut | Travel | Total |
| PressLight | 1246 | 254.4 | 252.0 | 4866 | 219.6 | 362.8 | 1355 | 560.3 | 930.3 |
| CoLight | 1248 | 222.3 | 250.9 | 4986 | 259.5 | 374.8 | 2589 | 318.9 | 311.3 |
| 2dimRL | 1282 | 237.9 | 243.4 | 5010 | 252.4 | 376.3 | 2574 | 328.1 | 322.6 |
| FairShare(T) | 1287 | 251.3 | 243.4 | 4976 | 244.0 | 377.0 | 2561 | 331.2 | 330.1 |
| Decision(T) | 1292 | 224.6 | 239.7 | 5081 | 251.3 | 359.1 | 2586 | 327.6 | 318.6 |
| FairShare(TS) | 1285 | 251.6 | 243.7 | 5137 | 239.9 | 343.8 | 2583 | 327.3 | 318.1 |
| Decision(TS) | 1298 | 186.4 | 234.5 | 5186 | *277.4* | 357.8 | 2586 | 325.5 | 316.2 |

Table 3: Performance of Goodness EcoLight for worst case (fairness) metrics for 16x3

| Model | WrstTime | WrstWait | MaxWait | Stuck75 | Stuck50 | Stuck25 | Stuck0 |
|---|---|---|---|---|---|---|---|
| Presslight | 3516.4 | 2481.5 | 255.6 | 99.2 | 338.5 | 843.9 | 1405.9 |
| Colight | 834.4 | 900.8 | 45.9 | 0.0 | 0.4 | 2.7 | 234.7 |
| 2dimRL | 985.3 | 1396.5 | 47.6 | 0.9 | 2.4 | 6.0 | 250.1 |
| FairShare(T) | 1207.1 | 1524.2 | 48.7 | 2.1 | 6.2 | 14.1 | 261.3 |
| Decision(T) | 924.7 | 1352.2 | 47.7 | 0.0 | 0.0 | 1.4 | 238.3 |
| FairShare(TS) | 929.0 | 1320.0 | 48.5 | 0.0 | 0.0 | 0.5 | 241.0 |
| Decision(TS) | 675.2 | 1007.0 | 46.8 | 0.0 | 0.0 | 0.0 | 237.6 |

average wait times at any intersection) and (d) StuckX (vehicles stuck in network at X% time from simulation end). Fairness loss due to quantization is not only gracefully recovered by our goodness metrics, but we significantly outperform all baselines as well.

Using a finite sized LUT with (a) quantized traffic density values as rows and columns, and (b) cells containing binary decisions learnt using DRL model training, and model selection based on some goodness metrics, gives us performance and fairness comparable to the state-of-the-art DRL algorithms. This is extremely encouraging in terms of practical deployment in developing countries.

## 5 Doing Away With Look-up Tables: Threshold based Intersection Control

Based on anecdotal discussions with intersection control companies, while most intersections in developing regions will be able to support LUTs, some intersections might be budget constrained to such an extent that the controller's RAM will not be enough to even store LUTs. In this section, we therefore consider how to design such a stateless controller, with better performance and fairness metrics compared to other widely deployed stateless controllers. We start by examining the 1dim RL tried in Section 3, and gradually build performant and fair stateless control.

1-dimensional state RL did poorly on the Throughput and TotalTime metrics in Table 1, especially for the 16x3 road network. Wondering what is being learnt by the RL for the case of 1-dimensional state, we checked the model behaviour for the whole range of this state variable $< x3 = x1/(x1 + x2) >$ from 0.0 to 1.0. We calculate the expected value of signal change for all 16 intersections (of 16x1 NY road network) for continuous 50 rounds after training for 500 rounds.

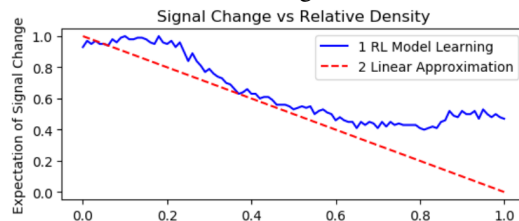

Figure 4: Density vs action

Figure 4 plots the expected signal change along y-axis, with relative density along x-axis. The signal change expectation is high when relative density is low (top left) and vice-versa (red line given for reference for exact negative correlation between signal change expectation and relative density). The blue curve shows a near-linear response following the red line, but is still non-linear. Thus 1-dimensional state with ratio $x1/(x1 + x2)$ is not enough to capture the necessary non-linearity and overall traffic concentration - empty vs. moderate vs. saturation. It only captures relative density among approaches, while absolute values retained in 2-dimensional state RL are clearly important.

We explore the options of both 1-dimensional relative density $< x1/(x1 + x2) >$ and 2-dimensional absolute densities $< x1, x2 >$ in the simple algorithm below. The algorithm does not use any LUT to store the signal switching decisions learnt by RL for all possible states. It only uses few empirically learned thresholds. This is to support embedded hardware, that cannot use LUTs due to RAM constraints and would need the control algorithm to be completely stateless, possibly using only a few thresholding parameters.

**GetNextAction** ($cur\_phase, phase\_time, density\_list$)**:**
   $action \leftarrow 0$
   $total\_density \leftarrow sum(density\_list)$
   **if** $total\_density > 0$ and $phase\_time \geq CONFIG[MinGreen]$ **then**
      $relative\_density \leftarrow density\_list[cur\_phase]/total\_density$
      **if** $relative\_density < CONFIG[\alpha]$ **then**
         **if** $CONFIG[Mode]$ $is$ $Random$ **then**
            $ratio \leftarrow random(0.0, 1.0)$
         **else**
            $cycleTime \leftarrow CONFIG[CycleTime]$
            **if** $CONFIG[Mode]$ $is$ $Timed(2dim)$ **then**
               $cycleTime \leftarrow cycleTime \times total\_density \times 2/CONFIG[MaxDensity]$
               $cycleTime \leftarrow MAX(CONFIG[MinGreen], cycleTime)$
            **end if**
            $ratio \leftarrow phase\_time/cycleTime$
         **end if**
         **if** $ratio > relative\_density$ **then**
            $action \leftarrow 1$
         **end if**
      **end if**
   **end if**
   **return** $action$

The intuition behind the algorithm is (a) to take the CycleTime (i.e. the cumulative duration of all phases), and divide it among phases in proportion to their relative densities and (b) to increase CycleTime based on increasing absolute densities. At each decision making point, the agent allows the green signal to continue until the relative density for that approach has not fallen below a threshold $\alpha$. Below $\alpha$, signal can be switched. When CycleTime is defined (we call this variant Timed), the agent uses it in proportion to the relative density (Timed (1dim)), with optionally increasing the given CycleTime in response to absolute densities (Timed (2dim)). When CycleTime is undefined (we call this variant Random), it would switch randomly, but still proportional to the relative density.

<table>
<tr><td colspan="2">Table 4: Algorithm Hyper Parameters</td><td colspan="2">Table 5: Empirically Learnt Values</td></tr>
<tr><td>Param</td><td>Description</td><td>Algorithm</td><td>Properties</td></tr>
<tr><td>$\alpha$</td><td>Hold green above this threshold</td><td>FixedTiming</td><td>20s Min/Max Green</td></tr>
<tr><td>MinGreen</td><td>Minimum green per phase</td><td>MaxPressure</td><td>5s Min Green</td></tr>
<tr><td>CycleTime</td><td>Total green time over phases</td><td>SOTL</td><td>2/4 veh, 5s Min Green</td></tr>
<tr><td>MaxDensity</td><td>Maximum density at intersection</td><td>Random</td><td>$\alpha$=0.17, 5s Min Green</td></tr>
<tr><td>Mode</td><td>Random / Timed(1dim or 2dim)</td><td>Timed</td><td>$\alpha$=0.17, 150s Cycle</td></tr>
</table>

We compare the performance of our stateless algorithms against below baselines. These baselines also do not use any state, but work with few parameters as listed in Table 5. State-of-the-art research based RL methods like Presslight and CoLight are still in literature and not adopted in the real world. So these simpler baselines are the widely deployed intersection control algorithms across the world. Developing countries, typically, still use Fixed Timing signals.

**(a) Fixed Timing:** Signal switches in cyclic order to the next approach after fixed time intervals.
**(b) Max Pressure:** Pressure is calculated by the difference of vehicles on the incoming and outgoing lanes for the possible movements in each phase (Varaiya [2013]). Signal is switched to the phase with maximum pressure. If current phase pressure is not the maximum, we switch to the next phase.
**(c) Self-Organizing Traffic Light (SOTL):** This is a vehicle actuated mechanism (Cools et al. [2006]). There is a minimum phase duration. Once the minimum phase duration is over, the switch signal is generated if the traffic in green approach is less than a threshold and traffic in any other approach is more than another threshold.

Table 6: Performance of EcoLight Thresholding Algorithms for average case metrics

| | 1x1 | | | 16x1 | | | 16x3 | | |
|---|---|---|---|---|---|---|---|---|---|
| Algo | nOut | Travel | Total | nOut | Travel | Total | nOut | Travel | Total |
| FixedTiming | 1249 | 260.6 | 252.0 | 3743 | *193.0* | 583.5 | 1489 | 723.2 | 985.7 |
| MaxPressure | 1160 | 280.8 | 272.8 | 4106 | *214.4* | 504.2 | 1840 | 649.7 | 768.8 |
| SOTL | 1305 | 246.7 | 239.0 | 4640 | *264.7* | 436.3 | 2462 | 485.9 | 465.1 |
| Random | 1361 | 231.6 | 224.8 | 5076 | *354.9* | 427.8 | 2540 | 378.5 | 364.9 |
| Timed(1dim) | 1358 | 231.7 | 255.4 | 5104 | *355.3* | 428.9 | 2516 | 380.6 | 368.3 |
| Timed(2dim) | 1358 | 231.7 | 255.4 | 5268 | *346.6* | 406.3 | 2553 | 375.2 | 361.7 |

Table 6 shows the average case metric values (a) nOut (number of vehicles cleared by the intersection), (b) Travel (time spent by cleared vehicles) and (c) Total (time spent by all vehicles). Our algorithms Random, Timed (1dim) and Timed (2dim), clear many more vehicles at lower Travel and Total times than the baselines, for all benchmark datasets. The Travel times for 16x1 network is higher (italicized in Table 6) for our algorithms, though other metrics improved. This is due to the fact that it is a linear network of 16 intersections and the traffic pattern is such that a good part of the traffic enters around one end and exits around the other (and vice-versa), making the vehicles cross many intersections in a sequence. Supported by increased nOut, our algorithms make more vehicles to exit the network. The extra vehicles which exit are mostly the ones with larger travel times, thus pushing the average travel time for all cleared vehicles higher. Similar behaviour is observed for the baselines as well, where SOTL Travel time (with more nOut) is higher than other baselines (with less nOut).

Table 7: Performance of EcoLight Thresholding Algo for worst case (fairness) metrics for 16x3

| Algo | WrstTime | WrstWait | MaxWait | Stuck75 | Stuck50 | Stuck25 | Stuck0 |
|---|---|---|---|---|---|---|---|
| FixedTiming | 3443 | 2671 | 255.6 | 82 | 348 | 741 | 1203 |
| MaxPressure | 3100 | 2347 | 261.6 | 11 | 154 | 449 | 942 |
| SOTL | 1229 | 2188 | 79.8 | 0 | 0 | 11 | 362 |
| Random | 841 | 526 | 56.1 | 0 | 0 | 0 | 284 |
| Timed(1dim) | 839 | 524 | 56.5 | 0 | 0 | 0 | 308 |
| Timed(2dim) | 719 | 516 | 54.2 | 0 | 0 | 0 | 271 |

We further show the value of worst case or fairness metrics for 16x3 benchmark dataset in Table 7. For our Random variant, we take average of 5 rounds of simulation. For all others, the results are consistent for every round. Our algorithms significantly outperform the baselines for all fairness metrics for 16x3 network, and also for other benchmarks (omitted here for space constraints).

Based on these results, in situations where running RL based control or maintaining LUTs are not feasible due to RAM constraints, our stateless algorithms can be deployed, vastly improving both performance and fairness metrics, compared to the currently deployed intersection control baselines.

Table 8 shows performance of EcoLight Algorithms on two addition datasets collected at different times on the same intersection in New Delhi (India).

Table 8: Performance on other 1x1 datasets

| | 2 | | | 3 | | |
|---|---|---|---|---|---|---|
| Algo | nOut | Travel | Total | nOut | Travel | Total |
| PressLight | 225 | 30.0 | 29.4 | 529 | 163.1 | 177.6 |
| Colight | 221 | 31.2 | 49.1 | 514 | 163.2 | 181.7 |
| 2dimRL | 225 | 30.4 | 29.9 | 540 | 182.3 | 178.6 |
| FairShare(T) | 225 | 31.3 | 30.9 | 517 | 194.5 | 189.5 |
| Decision(T) | 225 | 30.3 | 29.8 | 538 | 185.2 | 180.2 |
| FairShare(TS) | 225 | 32.2 | 31.7 | 538 | 181.7 | 178.4 |
| Decision(TS) | 225 | 30.4 | 29.9 | 564 | 172.8 | 167.8 |
| Timed(2dim) | 225 | 31.9 | 31.3 | 566 | 168.9 | 162.5 |
| Timed(1dim) | 225 | 31.9 | 31.3 | 563 | 174.8 | 166.9 |
| Random | 225 | 31.9 | 31.3 | 562 | 172.8 | 166.3 |
| SOTL | 223 | 50.7 | 53.8 | 539 | 196.0 | 186.7 |
| MaxPressure | 224 | 48.4 | 47.3 | 487 | 205.5 | 202.0 |
| FixedTiming | 224 | 70.9 | 69.0 | 512 | 198.4 | 188.8 |

# 6 Input to Control Algorithms: Computer Vision for End-To-End System

All intersection control algorithms designed in this paper – (a) 2dim and 1dim state DRLs (Section 3), (b) LUTs built from offline DRL training using quantized states (Section 4) and (c) stateless threshold based algorithms (Section 5), use traffic density as input. More specifically, the algorithms need density of standing traffic (also called stop density), discarding vehicles which have started moving.

Given the hardware constraints, we need to make sure that this input is available to our control algorithms at an acceptable latency, with limited computation and no communication to a back-end server. As efficient computer vision candidates, we use background subtraction and optical flow. A background filter is subtracted from each frame, to compute the foreground, and $foreground/background$ indicates traffic density. The filter is periodically updated, the period $\tau$ denoting the learning rate. Such updates ensure that changing lighting conditions over the day, shadows etc. are correctly incorporated in the background filter. Just like learning LUTs using computation heavy DRL training, here also $\tau$ is learnt using compute intensive offline analysis, namely CNNs for vehicle detection (Chauhan et al. [2019]). The CNN outputs vehicle bounding boxes on training videos. $\tau$ is empirically set, so that density estimates from background subtraction match the CNN bounding box detection based density estimation.

Background subtraction based density estimates comprise both standing and moving traffic, whereas the control algorithms need to discard density contributed by the moving vehicles. So we additionally use optical flow algorithm, to detect moving pixels between frames, and compute standing traffic density from the stationary parts of the frames.

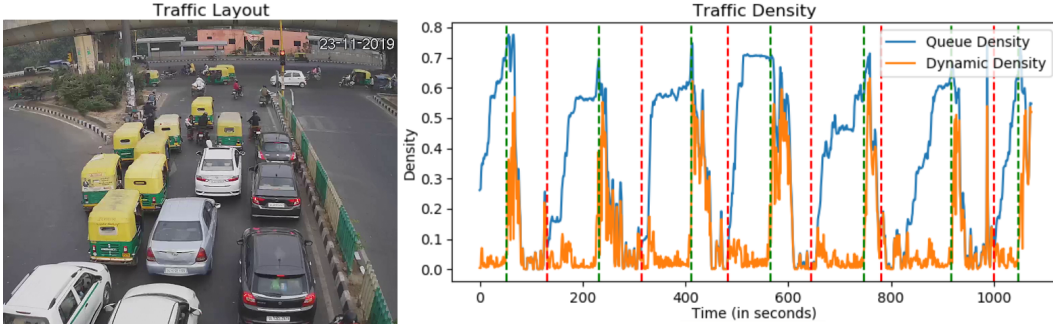

Figure 5: Developing Region Traffic Density Estimation

Figure 5 on the left shows a high traffic density frame, from one approach of a developing region intersection we are working at. The graph on the right shows for this location: (a) background subtraction based density (Queue Density in blue curve) and (b) optical flow based density (Dynamic Density in orange curve), over a span of over 15 minutes. Queue density starts to rise when signal turns red (indicated by vertical red lines), and starts to fall when signal turns green (indicated by vertical green lines). Dynamic density is zero when red signal is on (between red and green vertical lines) and rises when signal turns green and vehicles start moving. The difference between these two curves gives the density of standing vehicles, the input required by our control algorithms. The density estimation code runs at 5 FPS on low cost embedded platform (1.8 GHz Intel(R) Atom(TM) CPU D525 with 4 logical cores and 8GB RAM) budgeted by our deployment partners. With signal keep-change decisions taken every 5-10 seconds using LUT or threshold based control algorithms, this FPS is good enough to get inputs for all approaches.

# 7 Conclusion

This paper presents EcoLight, an end-to-end intersection control system, including computer vision inputs and signal control decisions. EcoLight is practically deployable at low cost and without network connectivity in developing countries, and optimizes both performance (throughout and travel time) and fairness (worst case waiting time) metrics for the intersection. EcoLight exploits state-of-the-art powerful machine learning methods like Deep Reinforcement Learning (DRL) and Convolutional Neural Network (CNN) run on powerful GPU servers, to learn efficient runtime control deployable on low cost constrained embedded devices. Our deployment partners, who are traffic control authorities in a developing country, are enthusiastic with these promising results. We are exploring how to use federated learning to populate LUTs at different intersections with distinct traffic flow patterns, and also analyze the possible gain in reducing congestion and vehicular emissions with EcoLight deployment at key intersections.

## Broader Impact

Traffic intersection management has changed significantly over time, starting from in-person control, to timed-policy control, actuated control, network-switch control and currently to AI based control. With the wake of AI, utilizing deep learning and RL, we see greater power to control the traffic automatically without human intervention. But what if resource constraints make state-of-the-art DRL methods impossible to deploy in the developing world? The heuristics we learn in this paper using offline AI/ML algorithms, greatly improve metric values over traditional control methods currently deployed in the real world.

We also demonstrate an end-to-end working system, that needed significant engineering and logistic efforts. But this boosted the confidence of our deployment partners, that we are more serious about this work than writing a couple of research papers. This paper, therefore, is an application of computer science methodologies, to the real world problem of traffic intersection control. Its potential impact on environment and sustainability overrides its academic contributions, which might feel like lacking novel contributions. However, exploiting the advances of CNN to learn better thresholds for low overhead computer vision methods, or DRL to populate LUTs to just look up at runtime, might be considered as novel and important optimizations towards building a practical, deployable system.

We are working closely with the traffic control authorities, in collaboration with whom the intersection camera in this paper was deployed and data collected. Thus the extreme budget constraints, network unavailability issues etc. are real, as conveyed by anecdotal discussions with these deployment partners. More importantly, developing region datasets are not easy to come by. So to aid better collaborative research and more testing of these control ideas, labeled datasets on traffic flow and code will be released on paper acceptance. Video data will be shared individually, based on requests and discussions, to ensure privacy of people and cars captured in the camera view.

In terms of safety concerns, if the traffic control system fails, it will have chaotic situation on the road, which may lead to human and mechanical injuries. So safety constraints, orthogonal to the control decisions, will be part of the deployed system. We also believe that our method will not be less stable than other state-of-the-art researches in the area and are constantly verifying our control decisions with the human experts (our deployment partners).

In terms of data bias, we of course could collect and use data from only one intersection for developing region. Even that was non-trivial, unless we showed some benefits in terms of travel time etc. (as we do in this paper). The promising results in this paper is a good first step in gaining our deployment partners' confidence, so that more data from different intersections, with possibly different traffic patterns, can be gathered in future (as much as research budget permits). This would remove bias, if any. We nonetheless use all open-source data that state-of-the-art DRL papers Wei et al. [2019a,b] experiment with, and match their performance. So our data/experiments are at least less biased than the state-of-the-art literature, which completely ignored developing region constraints.

## Acknowledgments and Disclosure of Funding

We (the students) would like to thank the faculty of our department for their insightful teaching in the courses undertaken. We would also like to thank Aabmatica Technologies Pvt Ltd (New Delhi) for their continuous support. They have enabled us for the real data collection, alongwith various experiments in-situ on the road, without which crucial analysis of the algorithms presented in this paper was not feasible.

## Footnotes

[1] https://www1.nyc.gov/site/tlc/about/tlc-trip-record-data.page

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
