[Reviews · NeurIPS 2020]

Review 1

Summary and Contributions: The paper presents algorithms for intersection control that can cope with the scarcity of compute resources and network bandwidth in developing countries. Instead of using previously published heavy Deep RL algorithms, the paper presents results for 1 or 2 dimensional RL, and then replaces the RL run by using lookup tables. It next presents a stateless algorithm for the case of RAM constraints, when the lookup table cannot be used. The evaluation of the algorithms is done both on public NYC data, considered in previous papers, and on data collected in one intersection in a developing country. That dataset had the additional challenge of not having lanes (the traffic density was analyzed using CNN and optical flow). The evaluation results show performance and fairness in the ballpark of the heavier algorithms. In some cases, the simpler algorithms presented in the paper have an advantage over the previous heavier ones.

Strengths: This paper considers both an interesting use-case that has a potential impact on society, and an interesting algorithmic problem. It presents non-trivial solutions, which were demonstrated in an actual deployment (a demo). The solutions address both the RL aspect of finding the right traffic lights policy and the computer vision aspect of analyzing the traffic density without lanes. I think this could be interesting for the NeurIPS audience.

Weaknesses: The paper could benefit from showing additional results for implementation in developing countries, which is the motivation for the algorithmic work. Currently results are provided for a single demo in a single intersection. Also, the data of that intersection is not provided (the authors state that it will be provided upon acceptance). The computer-vision methods used for analyzing the images are not described in detail, their code is not provided, and there is no analysis of the correctness of this specific part (just an end-to-end analysis). If this is not done well, it could make all the algorithms become similarly weak.

Correctness: The claims and methods seem correct, although there are some weaknesses, as described in the previous paragraph.

Clarity: The paper is well written.

Relation to Prior Work: Prior work was clearly discussed, including differences from the current work.

Reproducibility: No

Additional Feedback:


Review 2

Summary and Contributions: This paper studies intersection control under extremely low resource scenarios. The major contribution in my opinion is a (pilot phase) deployed end-to-end intersection control system that uses very limited computing resources. The main techniques being used are a combination of low-resource reinforcement learning and computer vision (to extract the input state end-to-end).

Strengths: 1. The system is deployed (pilot phase) in one intersection of a developing country. This has the potential of benefitting the society by exploiting the power of AI. 2. The problem setting where only extremely low resources are accessible is well motivated. The solution being given is well engineered and suits the target scenario. 3. The performances of the 3 major designs, namely state space reduction, off-line training and online execution and threshold-based policy have been demonstrated by simulation results. 4. The system also shows good robustness and fairness performances.

Weaknesses: 1. My main concern is that the paper, while being practical and useful in real-world, has limited technical contributions from an AI perspective. I do appreciate that the authors are trying to make the writing clear and simple, it somehow appears to be more like a collection of engineering efforts to make the system work. The paper is also written in a way that looks like a "technical report" rather than a machine learning type of paper. For example the Markov Decision Process underlying the problem is never formally defined, especially state transitions. It will be helpful to write down these definitions. 2. While the 2-d and 1-d state representation seems to work, there is no discussion as to "why" it works. What is the intuition behind this? Is it only restricted to the very simple settings (1*1 1*16 and 3*16) or is it generalizable to more complex problem scenarios? I would suggest the authors make a more in-depth discussion from this perspective.

Correctness: Yes.

Clarity: The paper is clearly written and easy to follow. Perhaps it could be presented in a more formal way with definitions of problem, and mathematical representations.

Relation to Prior Work: Yes.

Reproducibility: Yes

Additional Feedback: Thanks to the authors for clarifying my concerns in the rebuttal. My consideration of leaning toward rejecting the paper was based on the lack of technical contribution from an AI/ML perspective. Combing the rebuttal and other reviews I am convinced that it is an interesting direction for RL-based intersection control, and as mentioned by R5 and the rebuttal, is a promising field to "jump out of the pure application of traffic signal control problem". Therefore I will raise my score from 5 to 6. It will also be good for the authors to add a brief but formal description of the mathematic formulation of the problem.


Review 3

Summary and Contributions: This paper tackles the traffic signal control problem with the consideration of infrastructural limits in developed countries. They tried the following: 1. simplifying the state representations. 2. look-up table methods (RL might not be supported due to infrastructure budgets). 3.Threshold-based methods (lower budget than look-up table methods). Overall the paper is well written, with considerations well-explained, methods reasonably adopted.

Strengths: 1. Sound empirical evaluation. This paper uses all open-source data and makes the experiments easier to reproduce. Secondly, this paper compared with plenty of baselines, including RL-based and traditional transportation methods, and the results seem to match the state-of-the-art performance. 2. Simple methods with clear motivations. The recent advances in RL-based methods seem to overcomplicate, with different tricks proposed yet hard to reproduce for low-budget communities. This paper shows a clear message: sometimes we might not necessarily complicate the problem and simple methods like look-up tables can also beat state-of-the-art RL methods.

Weaknesses: 1. The techniques in this paper require manual design that might not of the current NeurIPS community. Currently, the community seems to prefer automated learning-based/optimization-based methods. Yet the look-up table and threshold methods proposed in this paper might not be appreciated by some researchers.

Correctness: Yes

Clarity: Yes

Relation to Prior Work: Yes

Reproducibility: Yes

Additional Feedback: I have a few suggestions that might make this paper more convincing: 1. Jump out of the pure application of traffic signal control problem. It is preferable to see how the message "simpler methods beats complicates methods with budget constraints" generalize to other problems, especially real-world applications like transportation or health care (other than games or robotics control). The community would much like to see how in the real-world cases the complicated methods fail generally and how to mitigate the failure.

[Author Response · NeurIPS 2020]

Regarding the submission 10836 to NeurIPS 2020, we are grateful to the reviewers for their time and efforts for understanding our work, appreciating the strengths, expressing their concerns and enlightening us with valuable suggestions. We further appreciate NeurIPS for allowing us to clarify on the concerns raised.

**Reviewer 2: Computer vision results not explicitly shown, also code-data release is missing:** We second this concern that any discrepancy in capturing the input from the environment will impact the performance. Since the functionality of YOLO was customized towards developing region traffic and evaluated in prior work (reference [3] in the paper), we verified our traffic density outputs using the results of [3] as ground truth. Also, code will be released for all modules, along with relevant data, upon acceptance as committed in the paper.

**Reviewer 2: Additional results from developing countries:** The extended results, that were held down due to space limitation, are given in Table 1. We presented the results of most heavy traffic data in the paper. We will try to include 1-2 more experiments, or at least mention that we have tested at the same intersection (due to logistic constraints of camera deployment), but at other times under different traffic conditions, and observed similar improvements.

Table 1: **Performance of EcoLight Algorithms on 1x1 Traffic intersection (at different times)**

| Algo | 2 | | | 3 | | | 4 | | | 5 | | |
|---|---|---|---|---|---|---|---|---|---|---|---|---|
| | nOut | Travel | Total | nOut | Travel | Total | nOut | Travel | Total | nOut | Travel | Total |
| PressLight | 1300 | 222.0 | 308.7 | 225 | 29.9 | 29.8 | 529 | 163.1 | 177.6 | 557 | 161.8 | 189.4 |
| Colight | 1394 | 207.9 | 280.5 | 221 | 33.1 | 51.2 | 514 | 163.2 | 181.7 | 514 | 159.0 | 208.2 |
| 2dimRL | 1418 | 276.3 | 277.8 | 225 | 30.8 | 30.4 | 540 | 182.3 | 178.6 | 574 | 182.4 | 189.0 |
| FairShare(T) | 1417 | 266.9 | 277.6 | 225 | 31.0 | 30.4 | 517 | 194.5 | 189.5 | 581 | 193.5 | 188.3 |
| Decision(T) | 1466 | 273.7 | 264.2 | 225 | 31.0 | 30.6 | 538 | 185.2 | 180.2 | 579 | 176.6 | 186.5 |
| FairShare(TS) | 1421 | 289.1 | 278.0 | 224 | 30.6 | 30.1 | 538 | 181.7 | 178.4 | 587 | 191.8 | 187.4 |
| Decision(TS) | 1487 | 266.1 | 256.5 | 225 | 30.6 | 30.0 | 564 | 172.8 | 167.8 | 584 | 158.7 | 183.2 |
| Timed(2dim) | 1496 | 266.6 | 257.3 | 225 | 31.7 | 31.2 | 566 | 168.9 | 162.5 | 603 | 179.7 | 177.8 |
| Timed(1dim) | 1476 | 268.6 | 258.8 | 225 | 31.7 | 31.2 | 563 | 174.8 | 166.9 | 598 | 184.5 | 181.5 |
| Random | 1476 | 270.0 | 260.4 | 225 | 31.7 | 31.2 | 562 | 172.8 | 166.3 | 559 | 183.5 | 182.5 |
| SOTL | 1462 | 279.4 | 269.4 | 223 | 50.8 | 53.9 | 539 | 196.0 | 186.7 | 580 | 199.2 | 192.2 |
| MaxPressure | 1248 | 334.0 | 319.8 | 224 | 46.6 | 45.6 | 487 | 205.5 | 202.0 | 509 | 225.6 | 216.7 |
| FixedTiming | 1380 | 302.3 | 289.1 | 224 | 67.9 | 66.1 | 512 | 198.4 | 188.8 | 550 | 202.7 | 194.0 |

**Reviewer 3: Formal definitions absent:** We intended to start our paper by defining our problem as MDP, as the reviewer suggested. It would be similar to Section 3 of reference [14] in the paper. Due to space limitation, we eased on this section considering it a known formulation in the domain, favouring other sections for effective space utilization. We request the reviewer to consider a trimmed version of section 3 of [14] that we will include in our paper, as a fulfillment of the suggestion. We cannot express the equations in this document as well, due to space constraints.

**Reviewer 3: Why do small states work? Is this generalizable to larger road networks?** We observe that high dimensional states sometimes cause a qualified DNN to overfit. Carefully designed small states exploiting domain knowledge, retaining just enough required information, exhibit both better fitting and better generalization i.e. allow DNN to hold better understanding of the environment. We have analyzed this over a range of experiments, and the results are encouraging. Our work which carefully analyzes low dimensional states and what domain knowledge to use, is a research paper in its own capacity, and is awaiting review at a suitable conference. We avoided referring to that ARXIV-ed paper for anonymity reasons and will include a reference in this paper, if it is accepted. Hence, we started the given EcoLight paper with good confidence in low dimensional states. In terms of generalizability, all our methods are single intersection based, without any communication dependency, and thus should be generalizable to any number of intersections in large road networks, each intersection's agent working independently. This is reflected in our good results on competent open-source multi-intersection traffic data (16x1 and 16x3), also used in prior arts.

**Reviewer 5: Generalize optimized control for applications other than traffic:** For the LUT oriented work, the quantization was employed to facilitate efficient inference at runtime, despite of the repercussions it might cause to the performance, and our Goodness based RL methods effectively compensated for the same. These RL methods, independent of any quantization limitation, can also be utilized to improve fairness of any state-of-the-art RL/DNN design, exploiting the beauty of a sound heuristic. The enthusiastic reader-researchers would be able to deploy either/both of these independent features as per their requirements. We will also explore these other application domains, in context of resource constrained environments, as part of our future work. We will mention the same in the paper.

We acknowledge that due to limited space, we could not express everything we intended to. We had to sacrifice some background information in favour of experimental analysis and new contributions. If given an opportunity to restructure the paper (with/out increased space) after acceptance, we would love to incorporate your suggestions as best possible.

[Meta-Review · NeurIPS 2020]

This is an interesting paper on the application of ML / RL methods to the problem of reducing traffic congestion. The reviewers have come to a clear consensus that this paper is ready for acceptance at NeurIPS, and in discussion even the most critical reviewer has noted that this paper is ready for acceptance and that it would be a good contribution to the conference. I'd like to use this opportunity to ask the authors to ensure that they address the reviewer feedback fully in revision. In particular, there are several cases where the author response indicated "lack of space" as a reason why formal definitions, additional results, etc., were not included. Space is indeed difficult, but in cases like these preparing a supplement / appendix is a good way to go and can help the community learn as much as possible from the work. Please do address all feedback either in the main text or by creating a supplement or appendix. Many thanks!